# Microbiological Properties and Cytotoxicity of PNVCL Hydrogels Containing Flavonoids as Intracanal Medication for Endodontic Therapy

**DOI:** 10.3390/jfb13040305

**Published:** 2022-12-17

**Authors:** Gabriela Pacheco de Almeida Braga, Karina Sampaio Caiaffa, Jesse Augusto Pereira, Vanessa Rodrigues dos Santos, Amanda Caselato Andolfatto Souza, Lucas da Silva Ribeiro, Emerson Rodrigues Camargo, Anuradha Prakki, Cristiane Duque

**Affiliations:** 1Department of Preventive and Restorative Dentistry, Araçatuba Dental School, São Paulo State University (UNESP), Araçatuba 16015-050, Brazil; 2Department of Chemistry, Federal University of São Carlos (UFSCar), São Carlos, São Paulo 13565-905, Brazil; 3Department of Clinical Sciences-Restorative Dentistry, Faculty of Dentistry, University of Toronto, Toronto, ON M5G 1G6, Canada

**Keywords:** flavonoids, hydrogel, biofilms, cytotoxicity

## Abstract

This study aimed to evaluate the cytotoxicity and microbiological properties of poly (N-vinylcaprolactam)—PNVCL hydrogels containing flavonoids as intracanal medication for endodontic therapy. Antimicrobial activity of ampelopsin (AMP), isoquercitrin and rutin was determined against *Enterococcus faecalis*, *Actinomyces israelii*, *Lactobacillus casei*, *Streptococcus mutans*, and *Fusobacterium nucleatum* by the microdilution method. After synthesis and characterization by rheology, PNVCL hydrogels were loaded with AMP and controls calcium hydroxide (CH) and chlorhexidine (CHX), and determined the compounds release profile. PNVCL+AMP, PNVCL+CH, PNVCL+CHX were evaluated on multi-species biofilms and analyzed by Scanning Electron Microscopy (SEM) and Confocal Laser Scanning Microscopy (CLSM). Cytotoxicity was determined after fibroblasts exposure to serial dilutions of AMP and PNVCL hydrogel extracts. AMP was effective against all of the bacteria tested, especially against *S. mutans*, *A. israelli* and *F. nucleatum*. SEM and CLSM analysis showed that PNVCL + AMP caused a significant decrease and disorganization of multi-species biofilms and reduction of intracanal viable cells, superior to the other groups. AMP affected fibroblast viability at concentrations above 0.125 mg/mL, and extracts of PNVCL+AMP showed low cytotoxicity. In conclusion, PNVCL containing AMP demonstrated cytocompatibility and potent effect against multi-species biofilms and could be potential intracanal medication for endodontic purposes.

## 1. Introduction

Core microbiome of dental root canal infections are composed most frequently by up to thirty microbial species belonging to five phyla: Firmicutes, Bacteroidetes, Proteobacteria, Actinobacteria and Fusobacteria [1]. In primary endodontic infections, the most abundant species are Gram-negative obligate anaerobic species, however, anaerobic facultative bacteria, including *Streptococcus*, *Actinomyces species*, *Lactobacillus species*, *Parvimona micra*, *Pseudoramibacter alactolyticus*, *Enterococcus faecalis* and other, predominate in post-treatment infections, demonstrating their resistance to conventional chemical-mechanical therapies [1,2]. *Fusobacterium nucleatum* has been also involved with the development of the severe forms of endodontic flare-ups [3]. Intracanal bacteria are generally organized in multi-species biofilms embedded in polymeric matrix and attached to the dentinal root canal walls. In those communities, bacteria develop mechanisms of cooperation and antibiotic resistance, improving their metabolism, production of toxins, and defense from other bacteria or the host. Intraradicular biofilms have been associated with longstanding pathologic processes, including cysts, abscesses, and granulomas; infections more complicated to eradicate [4].

In infected root canals, intracanal medication is needed to eliminate residual bacteria after canal instrumentation, to reduce inflammation of periapical tissues, to neutralize toxins and necrotic debris, and to prevent micro-leakage from the temporary filling [4]. Calcium hydroxide—Ca(OH)_2_ (CH) is considered the recommended intracanal medication due to its favorable antimicrobial action, anti-inflammatory properties, and ability to induce the deposition of mineralized tissue [5]. CH has been used in different vehicles (aqueous, viscous, and oily) which influences its biological and physico-mechanical properties, including release profile of calcium and hydroxyl ions and time of action [6]. Despite these advantages, studies have shown the presence of biofilm/bacteria in the root canals even weeks after filling with CH medication, with significant reduction of lipopolysaccharide endotoxin [7,8]. Chlorhexidine (CHX) is a cationic bisbisguanide with wide-range antibacterial effect against Gram-positive and Gram-negative bacteria, fungi, and virus [9]. In Dentistry, CHX is considered as the gold standard of oral antimicrobials for dental caries and gingivitis control and is currently used as an irrigant or intracanal medication alone or in combination with CH for endodontic purposes [9]. However, chlorhexidine has presented toxicity when in contact with different cell lines, even at low concentrations [10].

In immature permanent teeth with apical periodontitis, high levels of bacteria and the presence of virulent species cause severe inflammation, which interferes with tissue healing and root maturation [11]. Current studies have evaluated the biological protocols which promote significant level of disinfection and preserve the remaining cells present in the pulp tissue and/or in the Hertwig sheath, allowing natural tissue regeneration [11,12]. One of these protocols is based on the concept of pulp revascularization combining passive irrigation with low concentration of sodium hypochlorite (NaClO) with subsequent intracanal medication with a tri-antibiotic paste [13] or CH [14]. Although this protocol has shown some favorable clinical results with the promotion of periapical tissue repair [15,16], it is considered of low predictability. This is because root development can vary from 21% to 100%, and there are some controversies regarding the tissue formed in the root canal that would consist of connective tissue with cementum associated with residual bacteria [11].

In the search for new biological compounds, a variety of flavonoids, compounds derived from fruits, plants, and roots, have been investigated for their multiple pharmacological properties [17]. Ampelopsin (AMP, dihydromyricetin) is a flavanonol derived from myricetin present in vine tea with biological properties such as anticancer, antioxidant, anti-inflammatory and antimicrobial effects [18]. In addition, AMP has increased the activity of alkaline phosphatase, mineral deposition, and expression of specific genes of osteoblasts without presenting cytotoxicity and negative effects on cell proliferation [19,20]. Isoquercitrin (ISO) has presented several therapeutic properties, including anti-inflammatory, antioxidant, anti-allergic, anti-fungal, and antibacterial activities [21]. Rutin (RUT, quercetin-3-O-rutinoside), a biflavonoid, present in some vegetables and citrus fruits has showed antibiofilm effect [22], antioxidant, anti-inflammatory, anticancer and antiallergic properties [23]. The three flavonoids can be detected in vine tea (*Ampelopsis grossedentata* Hand.-Mazz), and although ampelopsin is the major bioactive flavonoid, rutin and isoquercitrin have been also identified in leaves and unripe fruits [18].

Natural or synthetic polymers have been used as drug carriers to promote the controlled release of compounds and extend their biological effect. Advantages of these carriers include reduction of doses or frequent exposure to antimicrobials, and the risk of bacterial resistance [24,25]. Poly (N-vinylcaprolactam) (PNVCL) is a thermosensitive hydrogel, which presents critical values of temperature for the conversion to the gel state at the physiological human body temperature and can be a suitable aqueous vehicle for drug delivery. PNVCL hydrogel can transition from a liquid to gel state when injected into the human body and return to liquid state when cooled (thermoreversibility), without additional chemical reactions or environmental treatments [26,27]. Furthermore, PNVCL increases the low solubility of some drugs in aqueous solution, facilitating drug delivery, in addition to minimizing associated side effects [28]. Considering the wide range of therapeutic effect of flavonoids and the potential of PNVCL hydrogels as carrier of biomolecules, this study aimed to evaluate the antibiofilm properties and cytotoxicity of PNVCL hydrogel containing flavonoid in comparison to PNVCL containing CH or CHX.

## 2. Materials and Methods

### 2.1. Preparation of Flavonoids and Controls

The flavonoids evaluated in the study were: ampelopsin (AMP, #42866), isoquercitrin (ISO, #17793) and rutin (RUT, #R5143), all obtained from Sigma-Aldrich (St. Louis, MO, USA). Flavonoid stock solutions were prepared in dimethyl sulfoxide—DMSO at 30 mg/mL. Calcium hydroxide PA (CH, Biodynamics, Ibiporã—PR/BR) at 1 mg/mL and chlorhexidine gluconate (CHX, Manipullis Pharmacy, Araçatuba—SP/BR) at 20 mg/mL were diluted in sterile deionized water as positive controls. All compounds were filtered in a 0.22 µm membrane filter and store at—20 °C. Appendix A shows the flavonoids and controls used in the study, their codes, synonyms, chemical structure, and molecular weight. All the following experiments described were performed in triplicate in three independent experiments (*n* = 9/group) [29,30].

### 2.2. Determination of Antimicrobial Activity

#### 2.2.1. Microbial Strains and Growth Conditions

The following standard strains were used: *Enterococcus faecalis* (ATCC 51299), *Actinomyces israelii* (ATCC 12102), *Lactobacillus casei* (IAL#523), *Streptococcus mutans* (ATCC 25175), and *Fusobacterium nucleatum* (NCTC 11326) provided by Oswaldo Cruz Foundation—FIOCRUZ, Rio de Janeiro, Brazil. Bacterial suspensions were prepared as described by Caiaffa et al. [29] from cultures grown in Brain Heart infusion Agar—BHIA (Difco Laboratories, Kansas City, MO, USA) for *E. faecalis* and *A. israelii*, MRS Rogosa (Difco) for *L. casei*, Mitis Salivarius Agar (Difco) with 0.2 U/mL bacitracin (Sigma-Aldrich) for *S. mutans* and BHI blood agar (Difco) containing 5 mg/mL of hemin, 5 mg/mL of menadione, 0.5% yeast extract powder (YE, Difco), and 5% defibrinated sheep blood for *F. nucleatum*. All bacteria were grown in an incubator at 37 °C and 5% CO_2_, except *F. nucleatum* which grown in an anaerobic chamber (AnaeroGen, Oxoid, Thermo Scientific, Waltham, MA, USA).

#### 2.2.2. Determination of Minimum Inhibitory Concentration and Minimum Bactericidal Concentration

Minimum Inhibitory Concentration (MIC) and Minimum Bactericidal Concentration (MBC) assays were performed as a screening for the selection of the flavonoid with the highest antimicrobial effect. These assays followed the criteria described by the Clinical and Laboratory Standard Institute M7-A9 [31] for microdilution method in bacteria with modifications [29,30]. Briefly, bacterial cultures were grown until reaching approximately the density—OD = 0.5, centrifuged and pellets were re-suspended in Mueller-Hinton broth—MH (Difco) and BHI broth containing hemin, menadione, and YE for *F. nucleatum*. The bacterial concentrations were adjusted to 1–5 × 10^5^ CFU/mL. In 96-wells microplates, the antimicrobial agents (AMP, ISO and RUT) and positive control CHX were serially diluted from 1 to 0.0001 mg/mL in sterile deionized water and incubated with the bacterial suspensions. The microplates were incubated at 37 °C for 24 h for all bacteria, except *F. nucleatum*, which was incubated for 48 h. Afterwards, cultures were stained with 0.01% resazurin (#R7017, Sigma-Aldrich) for 3 h and absorbances analyzed at 570 and 600 nm in a spectrophotometer (Biotek, Winooski, VT, USA). In addition, serial dilutions from cultures of MIC wells (no detectable growth by colorimetric method) and three previous wells were plated on MH Agar (MHA) for 24h or in BHI blood agar for *F. nucleatum* for 48h. After counting of viable bacteria, MBC was determined when the compounds killed more than 99% of the tested microorganisms, compared to the negative control without antimicrobial agents.

### 2.3. Cytotoxicity Assays

#### 2.3.1. Cell Culture

Fibroblast cells from NIH/3T3 (ATCC CRL-1658) were grown in Dulbecco’s Modified Eagle’s Medium (DMEM, Gibco, Grand Island, NY, USA) supplemented with 10% fetal bovine serum (FBS; Gibco) and 100 IU/mL penicillin, 100 µg/mL streptomycin and 2 mmol/L glutamine (Gibco BRL, Gaithersburg, MD, USA) in an incubator with 5% CO_2_ and 95% air at 37 °C. Cell cultures were subcultured every 48 h until reaching 80% confluence. When cells reached 80% confluence, they were detached with trypsin-EDTA (0.25% Trypsin-EDTA 1x, Gibco) for 5 min at 37 °C and stained with Trypan Blue (Sigma-Aldrich) for posterior counting in a Neubauer chamber using an Automatic Cell Counter (Bio-Rad, Hercules, CA, USA). Next, they were seeded at a cell density of 5 × 10^4^ cells/well in 96-wells microplates and incubated for 24 h for subsequent cytotoxicity assay [29,30].

#### 2.3.2. Determination of Cell Viability

The viability rate was evaluated via colorimetric resazurin assay. This method allows determining cell respiration (mitochondrial), considering the metabolic rate of cells. The cells were treated with AMP, CH and CHX (from 1 to 0.0004 mg/mL) prepared in DMEM. After 24 h of treatment, cells were washed with PBS and resazurin 70 µM (Sigma-Aldrich) in DMEM was added for 4 h. Plates were read at 570 and 600 nm in spectrophotometer (Biotek, Winooski, VT, USA). The final values were obtained by the subtractions between the absorbance values at both wavelengths, and they were converted into percentage of cell viability considering the growth in DMEM medium without antimicrobials as 100% and the means determined for each group [32,33].

### 2.4. Synthesis and Characterization of Hydrogels

Thermosensitive poly (N-vinylcaprolactam) (PNVCL) hydrogels were synthesized and previously characterized by FITR, NMR, rheological analysis, and ultrastructural images by Medeiros et al. [26], Sala et al. [27] and Parameswaran-Thankam et al. [28]. In this present study, rheological assays were conducted to confirm the thermoresponsive behavior of PNVCL, as described previously by Sala et al. [27]. Briefly, 5 g of N-vinylcaprolactam (NVLC) monomer were dissolved in 20 g of dimethyl sulfoxide (DMSO) and heated to 70 °C under N_2_ atmosphere. Then, 0.112 g of azobisisobutyronitrile (AIBN) was dissolved in 8.33 g of DMSO and added (in drops) to the reaction system. The system was kept under agitation for 4 h, and the resultant polymer was purified by dialysis against deionized water for 4 days. PNVCL was dried in an oven at 50 °C and stored at 4 °C before use. Rheological analysis was performed to evaluate the hydrogel formation at 37 °C. Solutions with a concentration of 20 wt% of PNVCL were chosen based on the work of Sala et al. [27], who studied the hydrogel formation at different concentrations. The tests were performed on an Anton-Paar (Modular Compact Rheometer, Graz, Austria) model MCR 302, equipped with parallel geometry plate with 25 mm of diameter and gap of 1.0 mm. The time sweep analysis over 10 min was performed using strain of 1.0% and frequency of 1 Hz. For antibiofilm and cytotoxicity assays, PNVCL hydrogel were loaded with AMP at 2.5 mg/mL, CH at 1 mg/mL and CHX at 0.5 mg/mL separately and incubated for 24 h at 5 °C for total solubilization to prepare the polymeric gel. Considering the loading efficiency of the compounds, it can be assumed that all compounds are captured in the PNVCL hydrogel, and the encapsulation efficiency is considered 100%.

### 2.5. Evaluation of Antimicrobial Agents Release in PNVCL Hydrogels

For release assays, AMP solution was prepared by adding 5 mg of AMP in 0.25 mL of DMSO, followed by the addition of 9.5 mL of distilled water, at a concentration of 500 ppm. The samples were prepared by adding 200 mg of polymer to 1 mL of drug solution. This mixture was left for 24 h at 5 °C for total solubilization. 300 µL of sample was incubated for 10 min in the thermostatic bath to form the hydrogel, followed by the addition of 1.8 mL of buffer. A temperature of 37 °C and two different buffers were used to control the release: phosphate buffer saline (pH 7.4) and glycine buffer (pH 10.0). At predetermined times, 1 mL of supernatant was withdrawn and replaced with fresh buffer. AMP aliquots were analyzed in a UV-vis spectrometer at 325 nm. The release of CHX and CH, both at 500 ppm, from PNVCL hydrogels was also evaluated. The CHX aliquots were analyzed in a UV-vis spectrometer at 255 nm. For CH, a colorimetric method was used to allow its quantification (Calcio Liquidform #90, Labtest, Lagoa Santa, MG, Brazil). 1 mL of arsenazo III was added to each aliquot, turning it into a purple solution and determination of calcium content was determined at a length of 600 nm in a UV-vis spectrometer. To evaluate the release mechanism of the compounds, the Korsmeyer-Peppas model was used. Points were used up to the ratio Mt/M < 0.6, where Mt is the cumulative quantity released and M is the total quantity released.

### 2.6. Analysis of Antibiofilm Activity

#### 2.6.1. Multispecies Biofilms Assays and Analysis by Scanning Electron Microscopy

*E. faecalis*, *A. israelii*, *S. mutans*, *L. casei* and *F. nucleatum* cultures were mixed in equal aliquots at the same concentration (1–5 × 10^3^ CFU mL^−1^) in BHI broth containing 1% glucose and inserted in 24 wells microplates [30]. After 1 week of growth in anaerobic conditions and change of culture media every 48 h, biofilms were washed twice with sterile saline solution, and PNVCL without antimicrobials; PNVCL + AMP at 2.5 mg/mL (10× the highest MIC); PNVCL + CH at 1 mg/mL (usual concentration for clinical application); PNVCL + CHX at 0.5 mg/mL (100× the highest MIC) were inserted into each well. The plates were incubated for 48 h at 37 °C in anaerobic conditions. Aliquots from all wells were plated on BHI agar and the plates incubated for 48 h for further counting of CFU/mL. Concomitantly, the same experiments were conducted in coverslips for microscopic analysis. The samples were dehydrated in 70% ethanol for 10 min, 95% ethanol for 10 min and 100% ethanol for 20 min and air-dried in a desiccator. Then, coverslips were attached to aluminum stubs, gold coated by sputtering and analyzed in a scanning electron microscope (SEM) (Leo, Cambridge, MA, USA) [34].

#### 2.6.2. Multi-Species Biofilm Assays on Dentin Tubules and Analysis by Confocal Laser Scanning Microscopy

This study was approved by local Animal Committee (Protocol: 00444-2020) and conducted in accordance with Ma et al. [35] and Santos et al. [30]. Briefly, roots from bovine incisors (n = 6/group) were separated from crowns at 1 mm below the cementum-enamel junction using a diamond disc (KG Sorensen D 91, Barueri, SP, Brazil) under water irrigation using a precision saw (IsoMet 1000, Buehler, Lake Bluff, IL, USA). Then, the roots were horizontally sectioned to obtain 4 mm cylindrical specimens. The root canal of each specimen was abraded with a # 6 wide drill, and the root was sectioned again in two cylindrical halves. These specimens were cleaned with 17% EDTA in ultrasonic bath and distilled water to remove the smear layer. Afterwards, they were autoclaved and fixed into microtubes with composite resin. Cultures of *E. faecalis*, *A. israelii*, *L. casei*, *S. mutans*, and *F. nucleatum* were mixed in equal volumes at 1–5 × 10^3^ CFU/mL concentration, centrifuged and resuspended in BHI broth supplemented with 1% glucose. Dentin specimens were contaminated with 0.5 mL of the bacterial cultures, centrifuged (1400, 2000, 3600 and 5600× *g* for 5 min each) and incubated at 37 °C for 2 weeks. The culture medium was changed every 48 h. After this period, dentin samples were washed with sterile water and inserted in a new micro tube containing 350 μL of the following treatments: PNVCL without antimicrobials, PNVCL loaded with AMP at 2.5 mg/mL, PNVCL loaded with CH at 1 mg/mL, PNVCL loaded with CHX at 0.5 mg/mL (positive control) and sterile water (negative control). After an initial shaking of 1 h, samples were incubated for 48 h at 37 °C. Subsequently, the samples were washed with distilled water, cut into two new halves, and stained with 100 µL of fluorescent LIVE/DEAD BacLight Bacterial Viability stain (L13152, Molecular Probes, Eugene, OR, USA) for longitudinal observation of the dead and live cells inside dentin tubules by Confocal Laser Scanning Microscopy—CLSM (Leica TCS SP5, Microsystems GmbH, Berg, Germany) [35]. The excitation/emission wavelengths were 480/500 nm for SYTO 9 and 490/635 nm for propidium iodide stains. Ten-micrometer-deep scans were obtained from three randomly selected places using CLSM software (Leica AF 2.7.3.9723, Leica Application Suite, Advanced Fluorescence). The quantification of the red fluorescence ratio (dead cells) in relation to green- and-red fluorescence (live and dead cells) was determined for each antimicrobial agent tested by Image J 1.48 (NIH, Bethesda, MA, USA) [30,36].

### 2.7. Cytotoxicity Assays with Hydrogel Extracts

PNVCL hydrogel were loaded with AMP at 2.5 mg/mL, CH at 1 mg/mL and CHX at 0.5 mg/mL separately and incubated for 24h at 5 °C for total solubilization. After 24 h, the samples at the liquid state were incubated overnight at 37 °C to acquire the gel state. Then, the DMEM culture medium was added over the hydrogel and incubated for 48 h and 7 days. The supernatant was harvested after 48 h and 7 days for sequential treatment. The cell viability rate was evaluated by colorimetry resazurin assay (Sigma-Aldrich), as previously described. Briefly, fibroblastic NIH/3T3 cells were seeded into the 96 well plates (1–5 × 10^4^ cells/well) and incubated for 24 h under standard cell culture conditions. After incubation, DMEM was removed, and the dilutions of the hydrogel’s extracts were added to the cells (from 1/2 to 1/64 dilutions). NHI/3T3 cultured in DMEM without any extract was used as the control. After the 24 h treatment, the extracts were removed and 200 μL of resazurin (70 μM) were added for incubation at 37 °C for 4 h. The plates were analyzed in a spectrophotometer (Biotek, Winooski, VT, USA) at 570 and 600 nm [30,32].

### 2.8. Statistical Analysis

Considering the amplitude of bacterial counts, the data were transformed in Log (CFU + 1/mL). The constant +1 was added because some results had a zero count. Data from cytocompatibility and microbiological assays were expressed in means/standard deviation and submitted to One-Way (groups) or Two-Way ANOVA (groups and concentrations) and Tukey tests. SPSS 19.0 software (SPSS Inc., Chicago, IL, USA) was used to run the statistical analysis, considering *p* < 0.05.

## 3. Results

### 3.1. Antimicrobial Activity

Among the microorganisms tested, the flavonoid with superior antibacterial activity was AMP with MIC values ranging 0.125 to 1 mg/mL and MBC between 0.25 and 1 mg/mL. ISO and RUT had antibacterial action only against F. nucleatum with MIC at 0.25 mg/mL and 0.5 mg/mL, respectively, and MBC at 1 mg/mL for both flavonoids (Table 1).

### 3.2. Cytotoxicity of AMP and Controls

The effect of AMP, CH and CHX on viability of NHI/3T3 cells can be seen on Figure 1. AMP at the concentrations above 0.25 mg/mL of ampelopsin affected cell viability. CH did not affect cell viability at any concentration tested. There was no statistical difference comparing AMP and CH from 0.125 mg/mL. CHX was toxic to cells at a concentration above 0.0078 mg/mL.

### 3.3. Rheological Analysis

Figure 2 represents the rheological analysis of PNVCL-L (20% *w*/*v*). The formation of a hydrogel is evaluated through the relationship of G′ (storage moduli) and G″ (loss moduli) as a function of temperature and time. At 25 °C the loss moduli were higher than the storage moduli (G” > G′) characterizing that at this temperature the hydrogel remains in the liquid state. Different from the temperature of 37 °C capable of keeping the hydrogel in a gel state due to G′ > G”.

### 3.4. Release Assay

Figure 3 shows the in vitro release of AMP, CHX and CH from PNVCL at pH 7 and pH 10. PNVCL hydrogel was able to control the output of the three compounds without showing a burst of release in the first hours. This property observed at the beginning of the test was maintained for the entire period of 7 days at neutral pH and at basic pH. The greater release of AMP and CH from PNVCL occurred at basic pH. The release of AMP increased from 13% to 37% at pH 10. For CH the increase occurred from 42% to 67% at the same pH. CHX had similar release patterns (20–25%) independent on the pH, indicating that the change in pH did not affect its release. Figure 3 also shows the calculated parameters and equation fitting to each release curve. K is the Peppas constant, and n is the exponential coefficient that indicates the release mechanism of the drugs. For values less than 0.5, the release is diffusion controlled. While for values between 0.5 and 1.0, the mechanism is anomalous transport, a mixture of polymer erosion and diffusion. At neutral pH, the diffusion of molecules into the supernatant is the determining step. However, at basic pH, both for AMP and CH, a change in mechanism occurs, becoming anomalous transport, indicating that the release of these compounds can be controlled by changing the pH. Furthermore, CHX presented *n* less than 0.5 in both cases, showing that its diffusion out of the hydrogel remains the determining step of the process.

### 3.5. SEM Analysis

Figure 4A–G shows representative SEM images of 7-days multispecies biofilms with E. faecalis, S. mutans, A. israelli, L. casei and F. nucleatum after treatment with the PNVCL hydrogels. PNVCL + AMP (Figure 4A) caused disorganization on multi-species biofilms, with evident reduction of bacterial cells and extracellular matrix, similar to that observed for the PNVCL + CH (Figure 4B) and PNVCL + CHX (Figure 4C) hydrogels. Pure PNVCL (Figure 4D) did not affect biofilm organization, as observed in the control group without any treatment (Figure 4E). Quantitative analysis of bacterial counts can be seen in Figure 4F. No statistical difference was observed between pure PNVCL and control group. A significant reduction in bacterial counts presented in multi-species biofilms was observed when AMP, CH and CHX were incorporated in the PNVCL hydrogel. No statistical difference between the PNVCL + AMP and PNVCL + CH group was observed, and PNVCL + CHX showed the highest reduction compared to the other groups.

### 3.6. CLSM Analysis

Figure 5A–F shows the results observed when 14-days multispecies biofilms (with *E. faecalis, S. mutans, A. israelli, L. casei and F. nucleatum*) formed inside root canals were exposed to the PNVCL hydrogels containing the antimicrobial agents for 48 h. Representative confocal images of PNVCL hydrogels are observed in Figure 5A–E. PNVCL + AMP (Figure 5A) showed a greater amount of dead cells (red dots), followed by PNVCL + CH (Figure 5B) and PNVCL+CHX (Figure 5C) compared to the control (Figure 5E) and PNVCL without antimicrobials (Figure 5D) that show more live cells (green dots). These results were confirmed in Figure 5F. PNVCL associated with ampelopsin reduced 73.8% of bacterial cells inside the tubules and had statistical difference compared to PNVCL + CH and PNVCL + CHX which reduced 58.8% and 50.6% of bacterial cells. There was no statistical difference between PNVCL + CH and PNVCL + CHX, however, both differed from the control. Pure PNVCL did not differ from the control group.

### 3.7. Cytotoxicity Tests with Hydrogels Extracts

Figure 6A–B shows the effect of the 48 h and 7 days extracts of PNVCL hydrogels containing AMP, CH and CHX, or without antimicrobials agents on fibroblast cells after 24 h of exposure. In Figure 6A, viability was superior to 70% when cells were exposed to 48 h PNVCL extracts and PNVCL containing AMP or CH with no statistical difference among them, at any dilution tested. Similar results were observed to PNVCL + CHX from dilutions below 1/8. In Figure 6B, 7-days extracts of PNVCL hydrogels containing AMP, CH and CHX did not statistically differ from each at any concentration and all of them were similar to control from dilution below 1/4. PNCVL + CHX showed no statistical difference with pure PNVCL at any concentration.

## 4. Discussion

Considering the challenge of developing a medication capable of substantially reducing residual bacteria in passive irrigation procedures and allowing the apex closure in immature permanent teeth, this study evaluated the antimicrobial activity of the flavonoids AMP, ISO and RUT. AMP had a higher bactericidal effect against the bacteria tested, especially against *A. israelli*, *S. mutans*, *F. nucleatum* with MIC/MBC ranging 0.125 to 1 mg/mL. Our study is according to previous study which observed that AMP was effective against Gram-positive bacteria (*Staphylococcus aureus* and *Bacillus subtilis*) and Gram-negative bacteria (*Escherichia coli*, *Salmonella paratyphi* and *Pseudomonas aeruginosa*) with MIC and MBC ranging 0.312 to 2.5 mg/mL [37]. Another study confirmed the antibacterial effect of AMP on *S. aureus* with MIC and MBC of 0.125 and 0.25 mg/mL, respectively [38]. The same study demonstrated a significant reduction of *S. aureus* biofilm biomass and a decrease in the metabolic activity of the biofilm cells. No study was found evaluating the effect of AMP on the same bacteria tested in the present study.

In general, hydroxylation of flavonoids improves the solubility and chemical stability of flavonoids and is also associated with their bioactive properties. Some investigators have confirmed a direct relationship between the antibacterial activity and the hydroxylation of flavonoids, mainly on the B and C rings [39,40]. Although hydroxylation at position 5 and 7 is presented in all flavonoids tested, AMP, ISO and RUT, AMP presents more hydroxyl groups on the B ring compared to the other ones. Besides, glycosylation at C3 position instead of hydroxylation, as observed in rutin and isoquercitrin, reduced their antibacterial effect (See Appendix A). Considering other mechanisms of action, AMP was capable of preventing cell division and proliferation, destroying cell integrity and permeability, in addition to interacting with bacterial DNA of *S. aureus*, possibly for its lipophilic characteristics [41]. Contrary to our study, some investigations demonstrated antimicrobial effect of RUT on *A. naeslundii*, *A. viscosus*, *C. albicans* [42], *E. coli* and methicillin-resistant *S. aureus* with MIC value ranging 0.4 to 1.6 mg/mL [22] and effect of ISO against *E. coli* and *C. albicans* with MIC value of 2.5 mg/mL [21]. Differences in results may be due to different methodologies and microorganisms tested. Based on the results obtained in this study, AMP was selected for the cytotoxicity assays.

In addition to the antimicrobial action, the compound used for endodontic purposes, should present compatibility with the remaining periapical cells, including fibroblasts, that will favor the tissue repair and allow the development of the root apex in case of immature teeth. In the present study, AMP did not affect fibroblast viability in the concentrations below 0.25 mg/mL. Studies have reported low cytotoxicity of AMP on fibroblasts and keratinocytes cells [43]. In an in vivo wound-healing study, AMP promoted proliferation of fibroblasts and fibrocytes, reduced the signs of inflammation, manifesting immunostimulatory and remodeling properties capable of accelerating wound repair when antimicrobial activities are no longer needed [40].

Despite the low toxicity and wide range of therapeutic application, flavonoids have some limitations like low solubility in water, easy degradation in highly acidic medium and low bioavailability [44]. To overcome these drawbacks, several new forms of delivery systems have been developed for the encapsulation of these compounds reducing their limitations and improving their therapeutic effect [44]. This is the first study which evaluate poly (n-vinylcaprolactam)—PNVCL—as thermosensitive polymer for the delivery of AMP. As demonstrated in a previous study demonstrated, when PNVCL reaches a temperature above its lower critical values of temperature (LCST), it changes from a liquid to a gel state [27]. PNVCL has demonstrated its ability to carry antibiotics, such as ciprofloxacin, and promoted a time-controlled release [28].

The ability of the hydrogel to release drugs is directly dependent on characteristics such as temperature, the portion of hydrophilic or hydrophobic groups in the copolymer and hydrophilicity/hydrophobicity of the drug [45]. Vibrational spectroscopy analysis in the infrared region showed that release rate in hydrogels is higher for hydrophilic than hydrophobic drugs [45]. The low water solubility of AMP may be an advantage, leading to a slower release of the drug release in PNVCL system and making longer the time of the drug action and reducing medication re-intake [46]. The influence of pH on drug release from PNVCL hydrogel systems was previously studied by Fallon et al. [47]. They observed that PNVCL exhibited a burst of drug release at pH 6.8, compared to lower release in an acidic environment at pH 2.2. In the present study, higher drug release was promoted by increased pH. CH presented the highest percentage of release from PNVCL when compared to AMP and CHX. This result can be explained by the alkalinity promoted by CH when it is dissolved (pH 12.6 in water solution), different from CHX, which is an acidic solution, with pH around 5.0. On the other hand, AMP in solution, has a pH ranging between 5.5 and 6.5, which promoted lower release at neutral pH (from 5 to 10%) and greater release at basic pH (from 13 to 37%). Previous study showed that nanocapsules containing AMP showed two release phases, one called burst phase when 44% of the drug were release up to 6 h, followed by a sustained release phase, where the remaining 56% of the drug were released progressively over time [48]. Ciprofloxacin incorporated in guar-graft-poly(N-vinylprolactam) hydrogels showed an initial burst release, followed by a controlled pattern of sustained release [28].

The effect of new antimicrobial agents on biofilms have been considered in the medical studies, since biofilms are implicated in the pathogenesis and chronicity of several bacterial infections, including endodontic infections, that are difficult to successfully eradicate with conventional antimicrobial therapies [49]. In this study, the antibiofilm effect of PNVCL associated with AMP, CH and CHX was evaluated by bacterial counts and analyzed by SEM and CLSM. The concentration of compounds incorporated in PNVCL hydrogels was chosen based on previous studies which have shown that lower concentrations of antimicrobials (lesser than 10× MIC) have no significant effect on polymicrobial biofilms [50,51], since biofilm cells are hundreds of times more resistant to antibacterial agents than planktonic cells [52,53]. In this study, bacterial counts and SEM images showed evident reduction of bacterial cells and extracellular matrix when multi-species biofilms were treated with PNVCL + AMP, PNVCL + CH and PNVCL + CHX. The same was observed for CLSM results, intracanal multi-species biofilms were also reduced by the three formulations of PNVCL, with superior effect for PNVCL + AMP. This is a relevant finding since one of the major limitations of intracanal medication is to reach residual bacteria in dentinal tubules, secondary canals, and isthmus after chemical-mechanical treatment. Our results are similar to those found to AMP-loaded (dihydromyricetin) nanocapsules at 1 mg/mL reduced 67% of the *P. aeruginosa* biofilm population in 96 h of treatment compared to 41.2% for free AMP and 44.8% for blank nanocapsules, showing sustained release of the drug and an additional antimicrobial effect of nanocapsules [48]. Other formulations of PNVCL hydrogels with pectin (poly(acrylamidoglycolic acid-co-vinylcaprolactam) and silver nanoparticles presented inhibitory effect against *B. subtilis* and *E. coli* [54] and copolymerization of N-vinylcaprolactam onto caboxymethylcellulose also showed high antibacterial activity against *S. aureus*, *Proteus vulgaris* and *S. typhi* [55].

In general, 48 h and 7 days PNVCL hydrogel extracts containing the antimicrobials AMP, CH and CHX caused low toxicity to fibroblasts evaluated in the present study. PNVCL hydrogels have been considered cytocompatible when exposed to Caco-2 and pulmonary Calu-3 cell lines. It is an advantage when compared to other hydrogels, such as poly(N-isopropylacrylaminde)—PNIPAM, which hydrolyzes producing toxic compounds [56]. In vitro tests showed that cell viability was greater than 90% when PNVCL when exposed to human mesenchymal stem cells and bovine chondrocytes for 24 h [27]. In addition, the cytotoxicity of poly(vinylcaprolactam)—PVCL-based microgels loaded with doxycycline on HK-2 cells were investigated after 48 h incubation and the hydrogel showed no cytotoxicity at the concentration range of 0.1–50 µg/mL, and about 90% of cells were viable at the high concentration of 100 µg/mL, which is considered biocompatibility and little cytotoxicity of PVCL [57]. In the present study, 48 h extract of PNVCL + CHX caused cytotoxicity in dilution above 1/8. CHX has presented cytotoxicity when directly exposed to cells even at low concentrations [58].

In traditional drug administration, high doses or repeated administration are generally used to reach the therapeutic effect, however, as a consequence, there is an increase in the systemic and local toxicity, in addition to a decrease in the general efficacy of the drug along the time. Hydrogels are noteworthy because they provide a controlled release system and contribute to the continuous and unsaturated release, increasing the time of action against microorganisms and efficacy [59]. In the present study, PNVCL demonstrated a controlled pattern of AMP release which promoted potent effect against multi-species biofilms associated with endodontic infections.

## 5. Conclusions

Among the flavonoids, AMP was the only effective against all bacteria tested, especially against *S. mutans*, *A. israelii* and *F. nucleatum*. When loaded in PNVCL hydrogels, AMP caused a significant decrease and disorganization of multi-species biofilms and reduction of intracanal viable cells, similar or superior to CH and CHX. In addition, extracts of PNVCL+AMP showed low cytotoxicity. Considering the aforementioned limitations of current endodontic materials, PNVCL associated with ampelopsin could be an interesting intracanal medication for endodontic treatment of immature permanent teeth. However, more studies are necessary to clarify the mechanism of AMP against bacteria in multi-species biofilms, at molecular level. Further in vivo clinical studies should be conducted to confirm the safety and efficacy of PVCL+AMP medication.

## Figures and Tables

**Figure 1 jfb-13-00305-f001:**
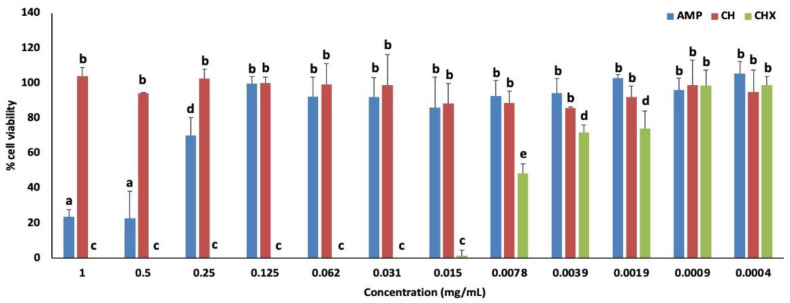
Effect of the flavonoid AMP and controls CH and CHX on the viability of fibroblasts (NIH/3T3) after 24 h of exposure, using staining with resazurin. Concentrations in mg/mL. a–e Different letters show statistical difference among all groups (comparing different antimicrobial agents at the same concentrations and comparing different concentrations of the same antimicrobial agent), according to Two-way ANOVA and Tukey test (*p* < 0.05).

**Figure 2 jfb-13-00305-f002:**
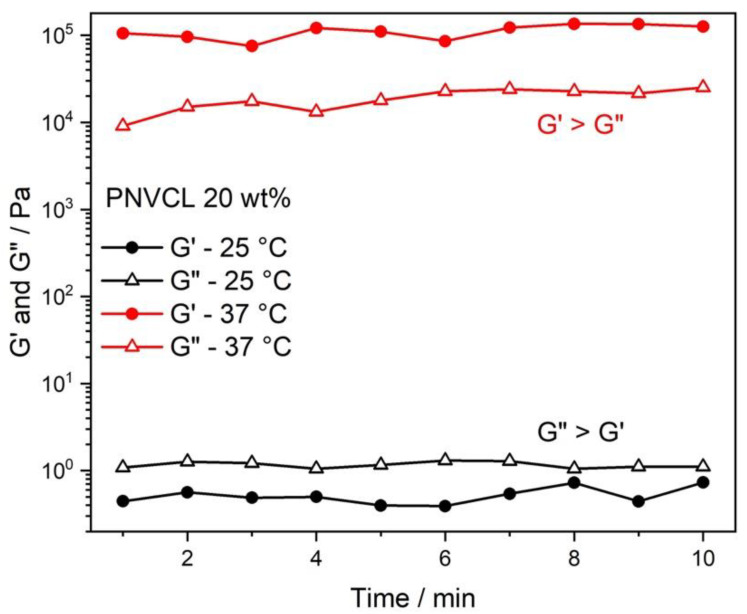
Variation of G′ and G″ with time for PNVCL—L (20% *w*/*v*) hydrogel at 25 °C and 37 °C, indicated by the temperature profile with time.

**Figure 3 jfb-13-00305-f003:**
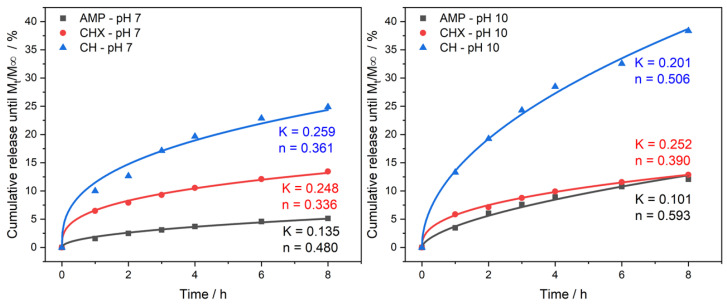
In vitro percentage of drug release vs. time profile for PNVCL hydrogels containing AMP, CHX and CH at pH 7 and pH 10. The mechanisms of drug release were also analyzed by the Korsmeyer-Peppas model. K represents the Peppas constant, and n is the exponential coefficient that indicates the release mechanism of the drugs. A basic pH, both for AMP and CH, a change in mechanism occurs, becoming anomalous transport, indicating that the release of these compounds can be controlled by changing the pH. Furthermore, CHX presented n less than 0.5 in both cases, showing that its diffusion out of the hydrogel remains the determining step of the process.

**Figure 4 jfb-13-00305-f004:**
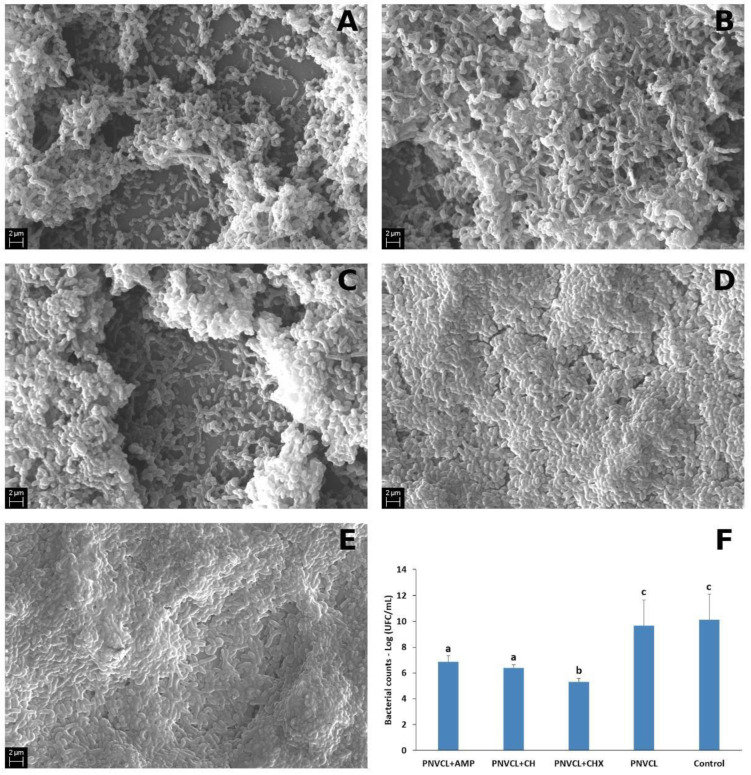
Representative SEM images of 7-days multi-species biofilms under 5000× magnification. Biofilms were treated for 48 h with (**A**) PNVCL containing AMP at 2.5 mg/mL; (**B**) PNVCL containing CH at 1 mg; (**C**) PNVCL containing CHX at 0.5 mg/mL; (**D**) PNVCL without antimicrobials; (**E**) Control—Bacterial growth without antimicrobial agents. (**F**) Mean (SD) of the bacterial counts detected after 48 h of the biofilm treatment with flavonoids and controls. a–c Different letters show statistical difference among the antimicrobial groups, according to One-Way ANOVA and Tukey test (*p* < 0.05).

**Figure 5 jfb-13-00305-f005:**
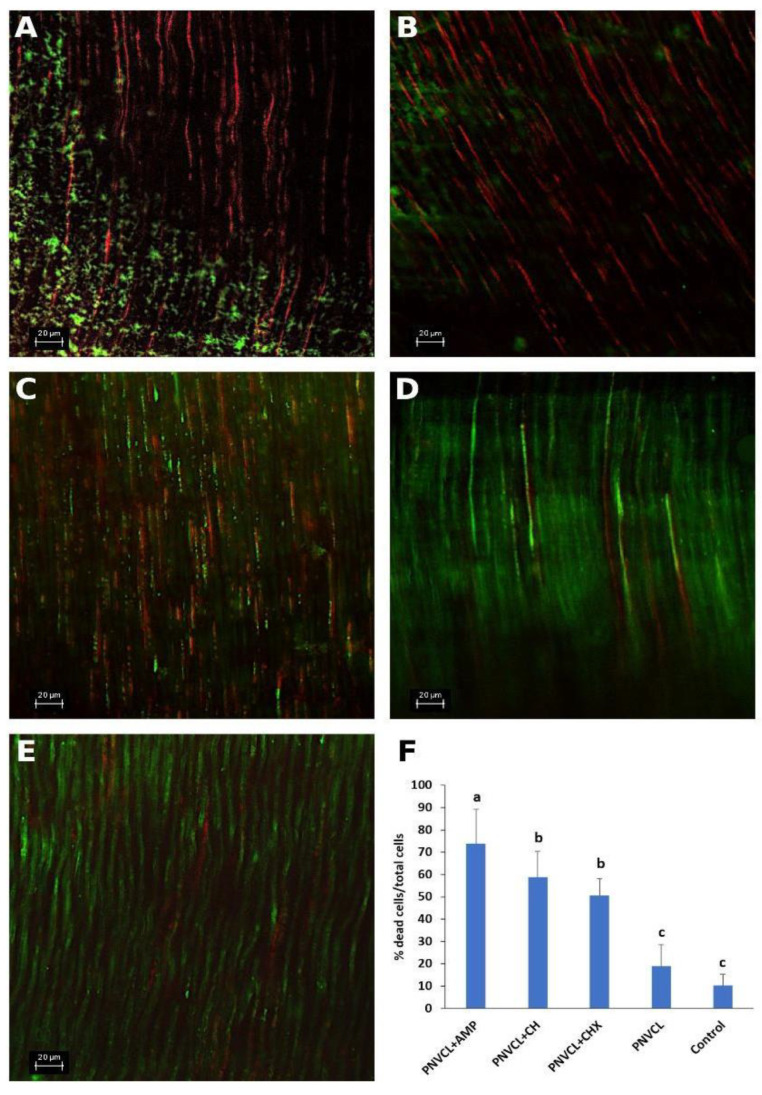
Representative confocal microscopy images of bovine root dentin specimens contaminated for 14 days with multi-species biofilms and treated for 48 h with the following groups with (**A**) PNVCL containing AMP at 2.5 mg/mL; (**B**) PNVCL containing CH at 1 mg/mL; (**C**) PNVCL containing CHX at 0.5 mg/mL; (**D**) PNVCL without antimicrobials; (**E**) Control—Bacterial growth without antimicrobial agents. (**F**) Mean (SD) of the percentages of dead cells of multi-species biofilms formed in root dentin of bovine teeth, after 48 h of treatment with antimicrobial agents and controls. a–c Different letters show statistical difference between the antimicrobial groups, according to One-Way ANOVA and Tukey test (*p* < 0.05).

**Figure 6 jfb-13-00305-f006:**
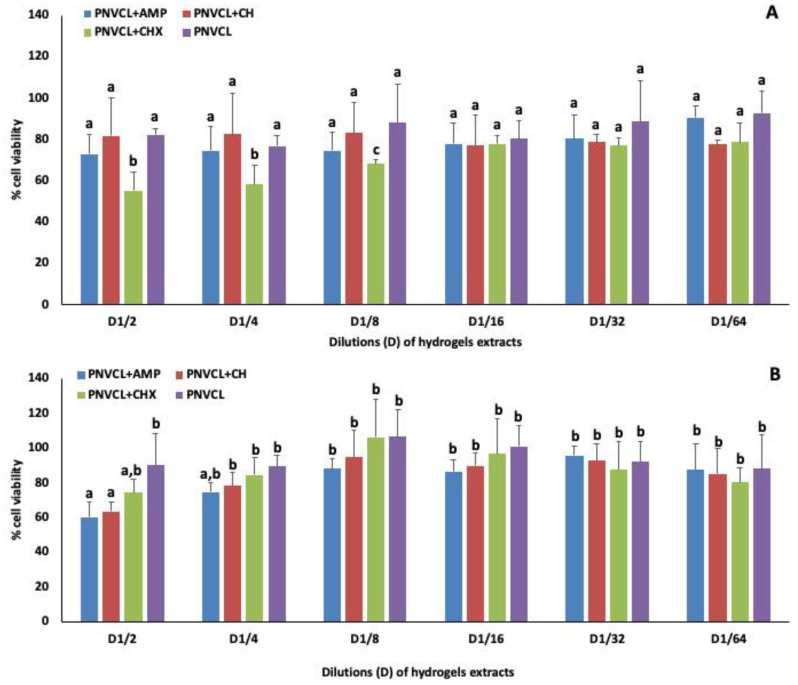
Effect of the 48 h (**A**) and 7 days (**B**) extracts of PNVCL hydrogels containing AMP, CH or CHX on the viability of fibroblasts (NIH/3T3) after 24h of exposure, using staining with resazurin. The extracts of PNVCL hydrogels were diluted from 1/2 to 1/64. a–c Different letters show statistical difference among all the antimicrobial groups (comparing different hydrogel extract at the same dilution and comparing different dilutions of the same hydrogel extract), according to Two-way ANOVA and Tukey test (*p* < 0.05).

**Table 1 jfb-13-00305-t001:** MIC and MBC (in parentheses) values in mg/mL for the flavonoids and control CHX against the oral bacteria tested.

	*E.* *faecalis*	*A.* *israelii*	*S.* *mutans*	*L.* *casei*	*F.* *nucleatum*
**AMP**	1 (1)	0.25 (1)	0.25 (1)	1 (1)	0.125 (0.25)
**ISO**	1 (>1)	1 (>1)	1 (>1)	1 (>1)	0.25 (1)
**RUT**	>1 (>1)	1 (>1)	1 (>1)	1 (>1)	0.5 (1)
**CHX**	0.005 (0.005)	0.002 (0.002)	0.0002 (0.002)	0.002 (0.004)	0.004 (0.004)

MBC: >99.9% cell reduction. MIC results were based on resazurin staining and MBC results were based on CFU/mL count in Miller Hinton Agar (MHA) medium. The growth of microorganisms without antimicrobials in MHA was considered 100%. The values of concentrations are in mg/mL.

## Data Availability

Not applicable.

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
