# Peer review of "Microbiological Properties and Cytotoxicity of PNVCL Hydrogels Containing Flavonoids as Intracanal Medication for Endodontic Therapy"

_jfb, 2022, doi:10.3390/jfb13040305_

Round 1

Reviewer 1 Report

The authors intended to assess the cytotoxicity and microbiological properties of PNVCL hydrogels containing flavonoids that are meant to be used for endodontic therapy after intracanal administration. In general, the study is very well designed and prepared. Here are only few points to consider to improve your manuscript:

Please, check the name of cell line 3T3-L1 throughout the manuscript.

It is better to specify the name of the method for determination of cell viability as colorimetric resazurin assay (not colorimetry) – line 161

Terms like „in vitro“ should be written in italics.

The manuscript lacks specification of the number of independent repetitions of the experiments.

It would be appropriate to specify in the figure legends the meaning of the different letters indicating statistical difference.

Reviewer 2 Report

Please find attached in word document.

Reviewer 3 Report

The manuscript “Microbiological properties and cytotoxicity of PNVCL hydrogels containing flavonoids as intracanal medication for endodontic therapy” by Almeida Braga1 et al. evaluates the cytotoxicity and microbiological properties of PNVCL hydrogels containing flavonoids as intracanal medication for endodontic therapy. Although, some preliminary results had been demonstrated, some important results are still lack from this manuscript. Therefore, I would suggest authors may take at least a major revision before publication. Here are the comments and suggestions:

1.     Table 1 mentioned on page 3 line 99 seems missing from this manuscript.

2.     Standard deviations should be added in Fig. 2.

3.     What would be the loading efficiency of these flavonoids?

4.     Some FTIR or NMR results should be added.

5.     In Fig. 3, what would be the mechanism of flavonoids released from these hydrogels.

6.     Both photos and SEM images of the PNVCL hydrogels with or without flavonoids should be provided.

7.     Finally, the conclusions should be extended.

Round 2

Reviewer 2 Report

I feel that the authors adequately addressed my comments. I do not think it is necessary to include empirical formulae in the Supplementary table. In addition AMP formula should have the 8 as a subscript. Otherwise I think this paper should be published.

Reviewer 3 Report

1.     The loading efficiency of these flavonoids seems still missing from this manuscript.

2.     Fig. 3 should be fitting with some release mechanism.

3.     NHI/3T3 should be corrected throughout this work.

Round 3

Reviewer 3 Report

It seems more acceptable now, although some typos (e.g. NHI) still found (line 274 and 300).